# Caloric Restriction Prevents Metabolic Dysfunction and the Changes in Hypothalamic Neuropeptides Associated with Obesity Independently of Dietary Fat Content in Rats

**DOI:** 10.3390/nu13072128

**Published:** 2021-06-22

**Authors:** Marina Martín, Amaia Rodríguez, Javier Gómez-Ambrosi, Beatriz Ramírez, Sara Becerril, Victoria Catalán, Miguel López, Carlos Diéguez, Gema Frühbeck, María A. Burrell

**Affiliations:** 1Department of Pathology, Anatomy and Physiology, University of Navarra, IdiSNA, 31008 Pamplona, Spain; mmartinr.1@unav.es; 2Metabolic Research Laboratory, Clínica Universidad de Navarra, IdiSNA, 31008 Pamplona, Spain; arodmur@unav.es (A.R.); jagomez@unav.es (J.G.-A.); bearamirez@unav.es (B.R.); sbecman@unav.es (S.B.); vcatalan@unav.es (V.C.); gfruhbeck@unav.es (G.F.); 3CIBER Fisiopatología de la Obesidad y Nutrición (CIBEROBN), Instituto de Salud Carlos III, 28029 Madrid, Spain; m.lopez@usc.es (M.L.); carlos.dieguez@usc.es (C.D.); 4Department of Physiology, CIMUS, University of Santiago de Compostela-Instituto de Investigación Sanitaria, 15782 Santiago de Compostela, Spain; 5Department of Endocrinology and Nutrition, Clínica Universidad de Navarra, 31008 Pamplona, Spain

**Keywords:** food restriction, gut hormones, hypothalamic neuropeptides and obesity

## Abstract

Energy restriction is a first therapy in the treatment of obesity, but the underlying biological mechanisms have not been completely clarified. We analyzed the effects of restriction of high-fat diet (HFD) on weight loss, circulating gut hormone levels and expression of hypothalamic neuropeptides. Ten-week-old male Wistar rats (*n* = 40) were randomly distributed into four groups: two fed ad libitum a normal diet (ND) (N group) or a HFD (H group) and two subjected to a 25% caloric restriction of ND (NR group) or HFD (HR group) for 9 weeks. A 25% restriction of HFD over 9 weeks leads to a 36% weight loss with regard to the group fed HFD ad libitum accompanied by normal values in adiposity index and food efficiency ratio (FER). This restriction also carried the normalization of NPY, AgRP and POMC hypothalamic mRNA expression, without changes in CART. Caloric restriction did not succeed in improving glucose homeostasis but reduced HFD-induced hyperinsulinemia. In conclusion, 25% restriction of HFD reduced adiposity and improved metabolism in experimental obesity, without changes in glycemia. Restriction of the HFD triggered the normalization of hypothalamic NPY, AgRP and POMC expression, as well as ghrelin and leptin levels.

## 1. Introduction

Obesity constitutes a persistent major health concern linked to increased morbidity and mortality [1]. Dietary intervention is still considered the cornerstone for the treatment of obesity and its associated metabolic alterations [2]. Many patients with obesity can achieve short-term weight reduction through diet alone, but successful long-term weight maintenance is much more difficult. The rise in obesity rates over the past 30 years has been related to increases in the portion size of energy-dense and highly palatable inexpensive food. Therefore, a reduction in portion size seems a logical alternative as a first therapeutic approach before progressing to a normal diet. Dietary interventions should be personalized, adapted to food preferences and enable flexible approaches to reduce calorie intake in order to strengthen motivation and adherence of patients with obesity [3]. In this regard, several control studies have shown that low-carbohydrate, high-fat diets (HFD) in patients with obesity and diabetes induce short-term favorable effects on weight loss, blood glucose and insulin [4]. Caloric restriction alleviates multiple complications of obesity and aging, including insulin resistance, dyslipidemia, hypertension, atherosclerosis and systemic inflammation [5,6]. However, it is not clear whether weight loss caused by a restricted HFD is accompanied by modifications in signals involved in the regulation of body weight and energy homeostasis.

Body weight is regulated by complex homeostatic mechanisms that include interactions between peripheral organs and the central nervous system [7]. In the hypothalamus, the arcuate nucleus (ARC) constitutes one of the main regulatory feeding centers. ARC neurons containing the orexigenic peptides neuropeptide Y (NPY) and agouti-related peptide (AgRP) and the anorexigenic factors proopiomelanocortin (POMC) and cocaine- and amphetamine-regulated transcript (CART) receive and integrate information about the metabolic state [8,9]. This integration occurs via receptors for hormones such as leptin, glucagon-like peptide 1 (GLP-1), peptide YY (PYY) and ghrelin [10], and also sensing the circulating levels of nutrients, such as glucose and fatty acids [11].

The hypothesis of the present study was that caloric restriction in rats fed a HFD might produce metabolic benefits. Therefore, we analyzed if body weight reduction derived from a restricted HFD is linked to modifications in glucose homeostasis, adipocyte size, gut hormone levels and hypothalamic neuropeptide expression.

## 2. Materials and Methods

### 2.1. Experimental Animals and Study Design

Ten-week-old male Wistar rats (*n* = 40) (breeding house of the University of Navarra) with a mean body weight of 308 ± 11 g were caged individually in a room under controlled temperature (22 ± 2 °C), relative humidity (50 ± 10%), ventilation (at least 15 complete changes of air/h) and a 12:12 h light–dark cycle (lights on at 8:00 a.m.). To analyze the effect of diet-induced obesity and caloric restriction, rats were divided into four dietary groups for 9 weeks (*n* = 10/group): rats fed ad libitum a ND (N group) (12.1 kJ: 4% fat, 82% carbohydrate and 14% protein, diet 2014S, Teklad Global Diets, Harlan, Barcelona, Spain) or a HFD (H group) (23.0 kJ/g: 59% fat, 27% carbohydrate and 14% protein, diet F3282; Bio-Serv, Frenchtown, NJ, USA) and rats fed a ND (NR group) or a HFD (HR group) with a caloric restriction of 25%. After 8 h fasting, rats were sacrificed by decapitation. Blood samples were immediately collected, and sera were obtained by cold centrifugation (4 °C) at 700× *g* for 15 min and stored at −20 °C. The perirenal, subcutaneous and epididymal white adipose tissue (WAT) depots were harvested, weighed and a small fragment of the fat tissues was fixed in 4% formaldehyde. The brain was also dissected out and frozen for study of the expression of hypothalamic neuropeptides. All experimental procedures were performed in accordance with the European Guidelines for the care and use of Laboratory Animals (directive 2010/63/EU) and were approved by the Ethical Committee for Animal Experimentation of the University of Navarra (049/10).

### 2.2. Body Weight, Body Composition and Food Efficiency Ratio

Body weight was recorded twice a week and food intake was monitored daily. The adiposity index was calculated as the sum of the weight of perirenal, subcutaneous and epididymal WAT depots in absolute (g) or relative (g/body weight) values. The food efficiency ratio (FER) was determined as body weight gained per week divided by total energy (kcal) consumed over this period.

### 2.3. Blood Measurements

Serum glucose concentrations were determined with a sensitive-automatic glucose sensor (Ascensia Elite, Bayer, Barcelona, Spain). Fasting serum leptin, PYY, GLP-1 and insulin concentrations were measured using a MILLIPLEX™ MAP rat gut hormone panel kit (RGT-88K Millipore Corporation, Billerica, MA, USA) in accordance with the manufacturer’s recommendations. Total ghrelin serum levels were assessed using a commercial ELISA kit (EZRGRT-91K, Millipore). Intra- and inter-assay coefficients of variation for measurements of total ghrelin were <5%.

### 2.4. Histological Analyses

Subcutaneous WAT samples were fixed in 4% formaldehyde, embedded in paraffin. Five µm-sections were stained with hematoxylin–eosin. Three fields per section from each animal were imaged with the 20× objective, and diameters from, at least, 100 adipocytes/section were determined with the Adiposoft software (version 1.13) plugin within ImageJ software (MATLAB).

### 2.5. In Situ Hybridization for Hypothalamic Neuropeptides

Cryostat coronal brain sections (16 µm) were obtained and stored at −80 °C until hybridization. In situ hybridization was carried out as earlier described [12]. Sections were probed with specific oligonucleotides for AgRP, CART, NPY and POMC (Table 1). These probes were 3′-end labeled with 35S-αdATP (Perkin Elmer, Waltham, MA, USA) using terminal deoxynucleotidyl transferase (New England Biolabs; Ipswich, MA, USA). The incubation of the sections with an excess of the unlabeled probes allowed to confirm the specificity of the probes. The frozen sections were treated with 4% paraformaldehyde in 0.1 mol/L phosphate buffer (pH 7.40) at room temperature (RT) for 30 min and then dehydrated using 70, 80, 90, 95% and absolute ethanol (5 min each). The hybridization was performed overnight at 37 °C in a moist chamber. Hybridization solution contained 0.5 × 10^6^ cpm (AgRP, CART and POMC) or 1 × 10^6^ (NPY) cpm per slide of the labeled probe, 4× saline-sodium citrate buffer (SSC), 50% deionized formamide, 1× Denhardt’s solution, 10% dextran sulfate and 10 µg/mL sheared, single-stranded salmon sperm DNA (all of them, Sigma-Aldrich; St. Louis, MO, USA). Then, the sections were washed in 1× SSC at RT, four times in 1× SSC at 42 °C (30 min per wash), one time in 1× SSC at RT for 1 h and then rinsed in water and ethanol. Finally, the sections were air-dried and exposed to Hyperfilm β-Max (KODAK; Rochester, NY, USA) at RT. All the slides were exposed under the same conditions and developed in developer/replenisher (Developer G150, AGFA HealthCare: Mortsel, Belgium) and Fixator (Manual Fixing G354; AGFA HealthCare: Mortsel, Belgium). Sections were scanned, and the hybridization signal was measured by densitometry using ImageJ-1.33 software (NIH; Bethesda, MD, USA). The optical density (OD) of the hybridization signal was quantified and corrected by the OD of its adjacent background value. A rectangle was outlined, always with the same dimensions, enclosing the hybridization signal over each nucleus and over adjacent brain areas of each section [12]. Sixteen to twenty sections for each animal (4–5 slides with 4 sections per slide) were used, and the mean was used as densitometry value for each animal.

### 2.6. Statistical Analysis

Data are shown as mean ± SEM. Differences between groups were analyzed by a two-way ANOVA (diet × caloric restriction) or a one-way ANOVA followed by the least significant difference (LSD) post hoc test, if an interaction was detected. The statistical analyses were performed using the SPSS v. 15.0 software (SPSS Inc., Chicago, IL, USA).

## 3. Results

### 3.1. Caloric Restriction Improved the Obese Phenotype even in Rats Fed a HFD

As shown in Figure 1A, all experimental groups exhibited similar body weight during the first 3–4 days of the experiment. Significant differences were identified from day 3 onwards. As expected, the H group showed the highest body weight (507 ± 18 g) at the end of the experiment, nearly doubling the initial one, while the NR group exhibited the lowest body mass (329 ± 4 g). Particularly relevant was the evolution of the HR group, which presented similar body weight to that of the N group throughout the experimental period (*p* = 0.165), being even lower at the end of the study. 

The percentage of body weight increase in each group during the 9 weeks of the experiment is illustrated in Figure 1B. As expected, groups fed a HFD showed a higher weight gain than those fed a normal diet (*p* < 0.0001) (Figure 1B) and higher relative and absolute adiposity indices (Figure 1C,D). Groups fed ad libitum experienced a more pronounced percentage of body weight increase than the corresponding groups subjected to caloric restriction (*p* < 0.0001). The comparison between N and HR groups was again noteworthy, since the percentage of body weight increase (*p* = 0.124) as well as absolute (*p* = 0.988) (Figure 1C) and relative (*p* = 0.721) (Figure 1D) adiposity index values were similar in both groups. 

The histological observation of subcutaneous WAT samples of the four groups revealed evident differences in adipocyte size (Figure 2). As expected, the H group presented the highest percentage of large adipocytes (diameter > 150 µm) (Figure 2C), while the NR group showed the highest percentage of small adipocytes (diameter < 150 µm) (Figure 2B). The WAT of the HR group exhibited hypertrophic adipocytes (diameter ≥ 250 µm) interspersed among a majority population of small adipocytes (diameter ≤ 50 µm) (Figure 2D).

### 3.2. Caloric Restriction Reduced Food Efficiency Independently of the Type of Diet

Since the ND and the HFD provide a different energy content, food intake is illustrated both as total grams of food consumed and the corresponding kilocalories ingested (Figure 3). The evolution of food intake in grams relative to body weight (Figure 3A,C) revealed higher food consumption in groups fed a normal diet with or without caloric restriction (*p* < 0.0001). However, groups fed a HFD ingested more calories (*p* < 0.0001) (Figure 3B,D) and exhibited a higher FER (*p* < 0.0001) (Figure 3C) than those fed a normal diet. No significant differences were observed in the total energy content of food ingested by N and HR groups (*p* = 0.916), nor in the FER values (*p* = 0.193).

### 3.3. Caloric Restriction Ameliorated Metabolic Profile even in Rats Fed a HFD

The general characteristics of the metabolic profile of experimental animals during the dietary interventions are summarized in Figure 4. Consumption of a HFD increased both glucose (*p* < 0.0001) and insulin (*p* < 0.0001) concentrations, while restriction diminished only insulin circulating levels (*p* < 0.0001) (Figure 4A,B). The N and HR groups had similar insulinemia (*p* = 0.614), but glycemia was higher in the HR group (*p* < 0.01) (Figure 4A,B). 

In accordance with the adiposity index data, leptinemia was augmented in the HFD and ad libitum fed groups (*p* < 0.001) (Figure 4C). Furthermore, as expected, consumption of HFD decreased ghrelin levels (*p* < 0.0001), while restriction increased them (*p* < 0.05) (Figure 4D). In the case of circulating GLP-1, no changes were observed between the four groups (Figure 4E). PYY serum concentrations were decreased in groups fed a HFD (*p* < 0.05) (Figure 4F). Interestingly, the N and HR groups showed similar leptinemia (*p* = 0.266) (Figure 4C) and ghrelinemia (*p* = 0.196) (Figure 4D), but different serum PYY levels, which were higher in the N group (*p* < 0.05) (Figure 4F).

### 3.4. Caloric Restriction Ameliorated Metabolic Profile even in Rats Fed a HFD

Expression of the hypothalamic neuropeptides controlling appetite, NPY, AgRP, POMC and CART was measured by in situ hybridization (Figure 5). Type of diet and restriction affected the orexigenic neuropeptides NPY and AgRP in a different manner. The effect of restriction on NPY expression depended on the type of diet (Figure 5A), so that restriction of the normal diet caused a decrease in NPY (*p* < 0.01). By contrast, restriction of the HFD increased NPY (*p* < 0.01). AgRP was decreased in the groups fed the HFD (*p* < 0.05) and increased in the groups fed the restricted diet, although there was only a marginal effect (*p* = 0.055) on both normal diet and HFD (Figure 5B). Interestingly, N and HR presented comparable expression of NPY, AgRP and POMC but not of CART. CART expression was lower in the groups fed the HFD (*p* < 0.05) and ad libitum (*p* < 0.0001) (Figure 5D).

## 4. Discussion

The significant global burden of overweight and obesity requires lifestyle strategies facilitating successful long-term body weight management. Dietary weight loss programs are mainly based on a decrease in fat or carbohydrate content in food, along with an important reduction in meal size [13]. Although this reduction often results in initial weight loss, patients with obesity often fail to maintain the treatment [14]. Numerous studies support the beneficial effects of food restriction, but only some of them have analyzed the results of restricting a HFD [15,16,17]. Moreover, the findings available are difficult to compare because these studies differ in numerous parameters (species/humans, duration of the experimental period or percentage of intake restriction). The present study demonstrates that a 25% restriction of a HFD for 9 weeks led to a similar body weight evolution and adiposity index than ad libitum intake of a normal diet. Other authors have concluded that animals fed a HFD with caloric content similar to the control group entail increased adiposity in the absence of significant changes in body weight [18] and that body composition is not normalized unless dietary fat is reduced [19]. However, our results regarding food intake clearly demonstrate that eating diets with the same caloric content but different fat amount leads to normalization of body weight and whole-body adiposity. Thus, dietary fat content is not the unique determinant of body fat when caloric intake is not excessive.

The intake of diets with a high fat content exerts deleterious effects on metabolism, especially on the glucose homeostasis. Previous studies indicate that energy restriction decreases plasma glucose and improves glucose tolerance and insulin sensitivity [20,21], even when restricting a HFD [22]. Our data revealed that the intake of a restricted HFD results in normal insulin values but higher glycemia, suggesting the persistence of insulin resistance, which is related to the type of diet rather than to the amount of weight loss or the body fat content.

The amelioration of glucose homeostasis is usually associated with modifications in adipose tissue morphology [23]. Adipocyte hyperplasia and hypertrophy are independent of body weight but correlate with insulin sensitivity [23,24,25]. Consistently, we observed evident differences in adipocyte size between groups fed the normal diet ad libitum and the restricted HFD. The HR group presented a heterogeneous population of adipocytes containing very large and very small adipocytes. These observations agree with other authors reporting that individuals with obesity whose fat depots are constituted predominantly by few large adipocytes exhibit higher glucose intolerance than those subjects with the same degree of obesity but with many small adipocytes [24]. Thus, increased adipocyte size can be considered an independent marker of insulin resistance and hyperleptinemia [26]. The underlying mechanisms responsible for the dimension of fat cells are complex and appear to be related to the dynamics of adipocyte storage and removal rate in different locations and metabolic situations [23,27,28]. Data from our study suggest that not only the distribution but also the morphology of WAT have to be considered when assessing glucose homeostasis and obesity.

The deregulation of glucose homeostasis caused by the intake of a HFD could also lead to modifications of some regulatory peptides, such as GLP-1 or PYY. In addition to improving glycemia [29], GLP-1 reduces appetite, thereby supporting its use in the treatment of obesity and its comorbidities [30,31]. In the present study, serum GLP-1 concentrations remained unaltered, which is in accordance with previous observations [32]. However, other authors suggest that over 11 weeks of intake modifications result in a progressive change in the concentration of circulating GLP-1 [33]. Therefore, it seems plausible that the duration of our dietary intervention is not enough to detect significant modifications in GLP-1. On the other hand, different studies have shown the relevance of PYY in the etiology of obesity and type 2 diabetes [34,35,36]. PYY acts in the hypothalamus to activate melanocortin neurons that affect insulin sensitivity [37]. Our data are in agreement with this observation, since HFD intake led to a significant decrease in PYY concentrations. When comparing the N and HR groups, PYY serum levels were lower in the HR group. The hypothesis that glycemic control would be improved with restricted HFD was not supported by the study findings. It seems that the hormonal profile that improves insulin secretion includes the elevation of both PYY and GLP-1 levels [38]. 

Ghrelin and leptin constitute crucial factors for the control of body composition and glucose homeostasis. Many studies have determined that the presence of specific macronutrients in the lumen of the gastrointestinal tract influences serum concentrations of different gastrointestinal hormones [39,40,41]. Global analysis of the influence of the type of diet and restriction yielded significant differences in ghrelin and leptin circulating levels. It is noteworthy that no significant differences were found when comparing the N and HR groups. Considering the parallel evolution in body weight and energy intake of both groups, it can be concluded that modifications in ghrelin and leptin are mainly influenced by body composition. In the case of leptin, this correlation is clear since WAT is the main source of leptin [42]. Reports of modifications in circulating ghrelin after feeding diets with different macronutrients are highly variable [43,44,45]. Although early ghrelin studies indicated its key role in the control of food intake [46,47,48], based on the present study, a direct correlation between ghrelin secretion and body composition can also be put forward. New insights indicate that induced ablation of ghrelin cells in adult mice does not decrease food intake and body weight, so ghrelin may not be so determinant in appetite control and body weight [49]. Thus, the ghrelin system may have evolved to play a role in protecting against starvation and hypoglycemia [50] as it seems to occur in the NR group.

To directly evaluate the metabolic effects of restricting a HFD in the ARC, we measured the expression of hypothalamic neuropeptides. The prototypic first-order neuronal targets of leptin, insulin and ghrelin action are the catabolic POMC/CART and the anabolic NPY/AgRP neurons. These neurons trigger opposing effects on energy balance and are reciprocally regulated by changes in energy stores. Adaptive responses to perturbations in body fat mass involve changes in the activity of NPY/AgRP and POMC/CART neurons in the ARC. Feeding a HFD produces a decrease in the expression of hypothalamic NPY and AgRP [7,51]. By contrast, restriction increases the expression of NPY/AgRP [52] and decreases the POMC/CART [53]. Our data are only partially in agreement with these general premises. First of all, the abnormal lower expression of NPY in the NR group could be explained by the fact that ghrelin induces orexigenic effects in free feeding conditions but has no effect in animals under negative energy balance conditions such as being chronically food-restricted [54]. The N and HR groups showed a similar expression of NPY, AgRP and POMC. This finding is not surprising given that both groups presented similar body weight, adiposity index, total intake of calories and FER. Nevertheless, the lower expression of CART in the HR group compared to the N group does not follow this trend. Some authors reported that CART affects several biological processes in both lipid and glucose homeostasis. Depending on the hormonal context, CART has insulin-like or insulin-antagonistic effects [55,56]. Consequently, lower CART expression in the HR group might be related to the deregulation of glucose homeostasis observed in this group. In line with our observations, several studies have reported incongruous correlations of hypothalamic expression of orexigenic factors NPY and AgRP as well as anorexigenic peptides POMC and CART under conditions of caloric restriction and/or obesogenic environment [57,58]. Interestingly, molecular profiling of hypothalamic neurons at a single-cell resolution has revealed molecularly distinct clusters of AgRP- and POMC-expressing neurons with potential divergent metabolic functions, hormonal regulation and response to dietary changes [59,60,61,62]. Due to the complexity of the regulation of these hypothalamic centers, further studies are required to unravel the mechanism through which different diets affect the expression of hypothalamic neuropeptides.

## 5. Conclusions

In conclusion, restricted HFD intake leads to a decrease in body weight, adiposity, FER, circulating ghrelin and leptin levels similar to those produced by a normal diet ad libitum. Furthermore, restricting HFD also normalizes the expression of NPY, AgRP and POMC in the ARC but does not improve glucose homeostasis. Altered levels of circulating PYY and CART mRNA expression in the ARC could be involved in glycemic deregulation. Restriction of a HFD may be used as an initial therapy in overweight and obese patients in order to achieve important improvements in energy status before progressing to a normal diet. The development of first step dietary interventions not involving a dramatic change in the habitual diet of overweight or obese people could represent an interesting alternative in the initial stages of obesity treatment. Further studies are required to investigate this possibility.

## Figures and Tables

**Figure 1 nutrients-13-02128-f001:**
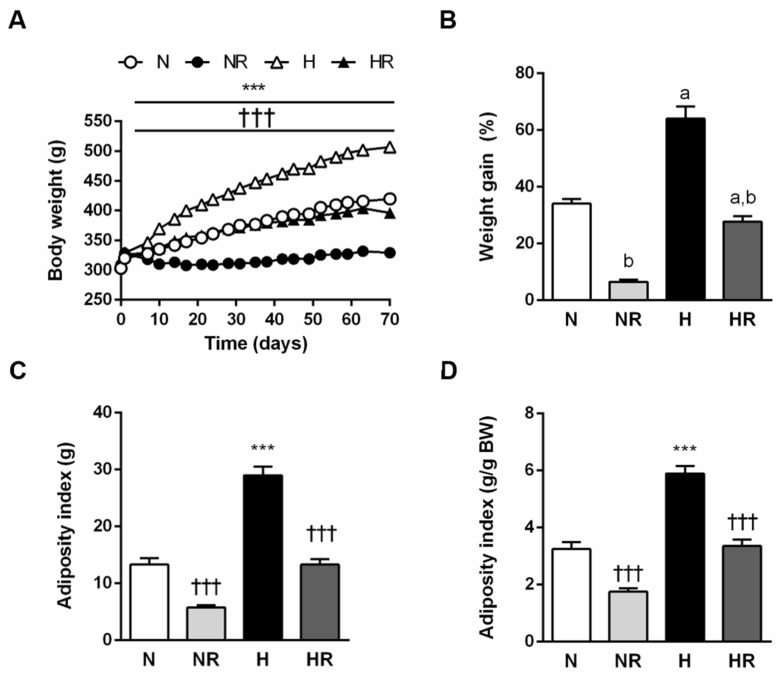
Growth curves of body weight (**A**), percentage of weight gain (**B**) and whole-body adiposity in absolute (**C**) and relative (**D**) values of the four experimental groups during the 9 weeks of dietary interventions. ^a^ *p* < 0.05 effect of diet; ^b^ *p* < 0.05 effect of caloric restriction. *** *p* < 0.001 vs. the same group fed a ND; ††† *p* < 0.001 vs. the same group fed ad libitum.

**Figure 2 nutrients-13-02128-f002:**
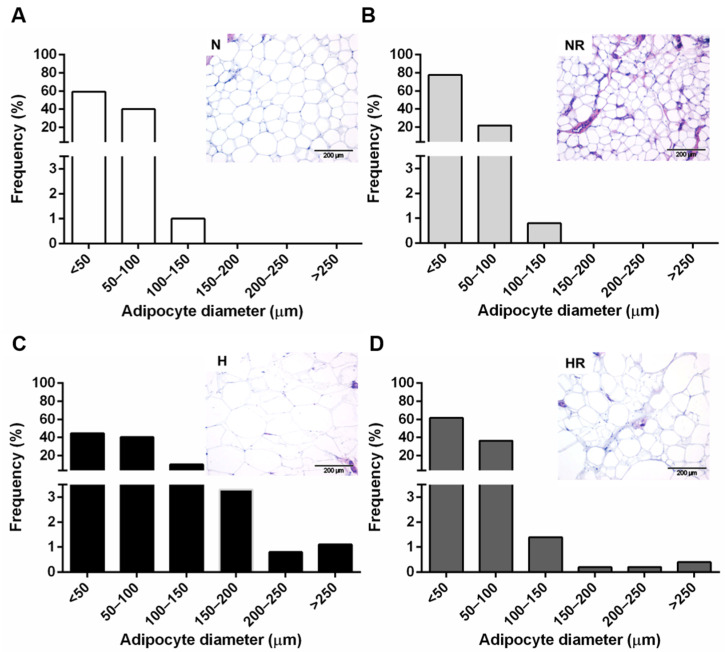
Adipocyte diameter distribution of subcutaneous WAT obtained from rats fed a normal diet (ND) (**A**) or a high-fat diet (HFD) ad libitum (**C**) or subjected to a 25% caloric restriction of ND (**B**) or HFD (**D**) for 9 weeks. Representative histological sections of subcutaneous WAT stained with hematoxylin–eosin are shown at the top of the histograms.

**Figure 3 nutrients-13-02128-f003:**
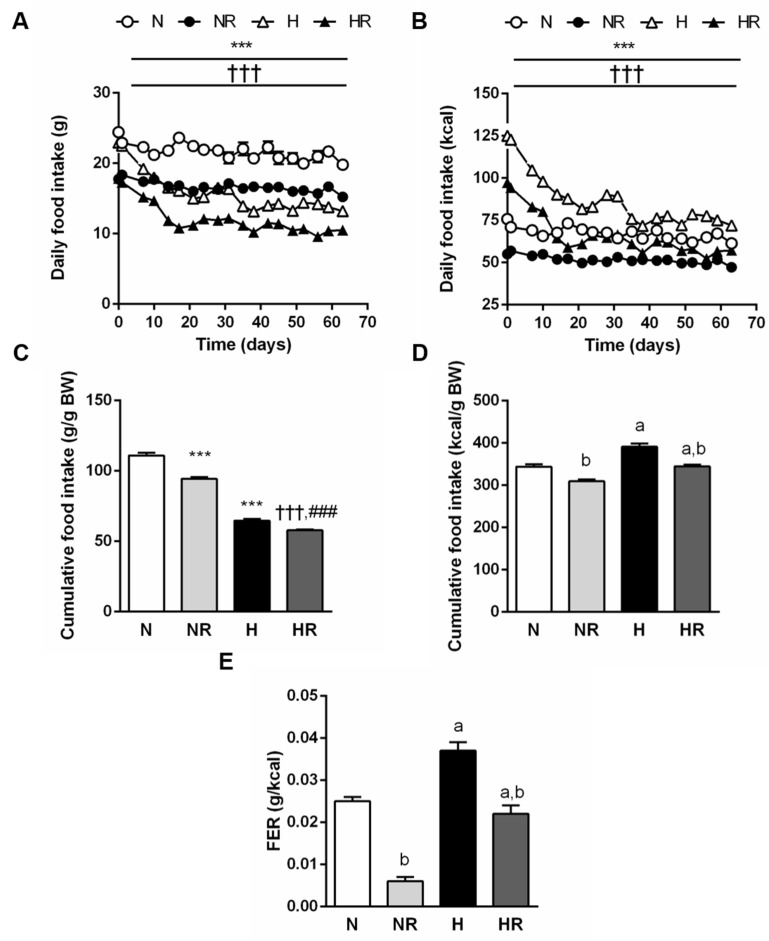
Curves of daily food intake in grams (**A**) and kilocalories (**B**) of the experimental animals (*n* = 10 per group). Bar graphs represent cumulative food intake relative to body weight in grams (**C**) and in kilocalories (**D**) as well as food efficiency ratio (FER) (**E**) during dietary interventions. ^a^ *p* < 0.05 effect of diet; ^b^ *p* < 0.05 effect of caloric restriction. *** *p* < 0.001 vs. the same group fed a ND; ††† *p* < 0.001 vs. the same group fed ad libitum; ### *p* < 0.001 vs. N.

**Figure 4 nutrients-13-02128-f004:**
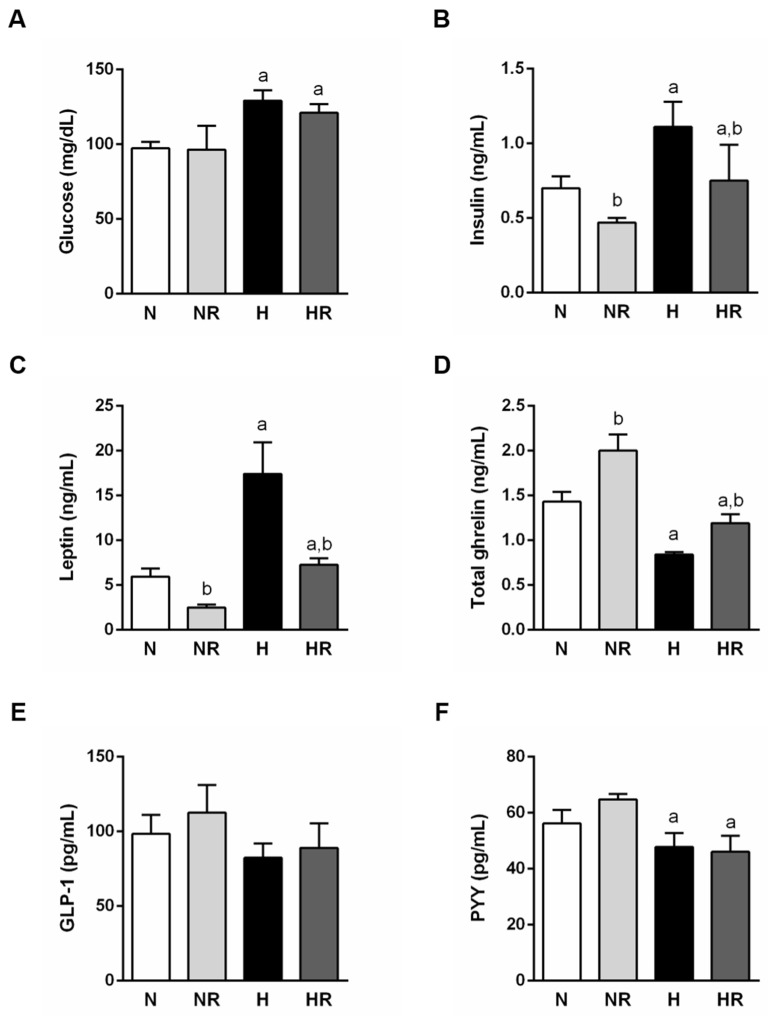
Fasting serum glucose (**A**), insulin (**B**), leptin (**C**), total ghrelin (**D**), GLP-1 (**E**) and PYY (**F**) levels of the four experimental groups. ^a^ *p* < 0.05 effect of diet; ^b^ *p* < 0.05 effect of caloric restriction.

**Figure 5 nutrients-13-02128-f005:**
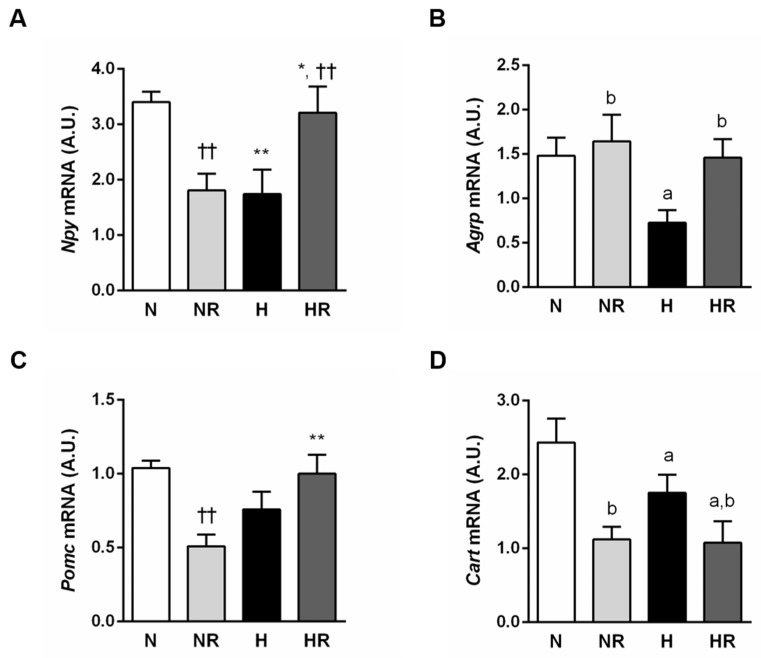
Hypothalamic gene expression of neuropeptides NPY (**A**), AgRP (**B**), POMC (**C**) and CART (**D**) in the arcuate nucleus of the four experimental groups. ^a^ *p* < 0.05 effect of diet; ^b^ *p* < 0.05 effect of caloric restriction. * *p* < 0.05; ** *p* < 0.01 vs. the same group fed a ND; †† *p* < 0.01 vs. the same group fed ad libitum.

**Table 1 nutrients-13-02128-t001:** List of in situ hybridization oligonucleotides.

Gene	GenBank ID	Sequence (5′-3′)
*Agrp*	NM_033650.1	CGACGCGGAGAACGAGACTCGCGGTTCTGTGGATCTAGCACCTCTGCC
*Cart*	NM_017110	CCGAAGGAGGCTGTCACCCCTTCACA
*Npy*	NM_012614.2	AGATGAGATGTGGGGGGAAACTAGGAAAAGTCAGGAGAGCAAGTTTCATT
*Pomc*	NM_139326	TCCATAGACGTGTGGAGCTG

*Agrp*, agouti-related neuropeptide; *Cart*, cocaine- and amphetamine-regulated transcript prepropeptide; *Npy*, neuropeptide Y; *Pomc*, proopiomelanocortin.

## Data Availability

The data presented in this study are available on request from the corresponding author. The data are not publicly available due to privacy restrictions.

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
