# Peer review of "Caloric Restriction Prevents Metabolic Dysfunction and the Changes in Hypothalamic Neuropeptides Associated with Obesity Independently of Dietary Fat Content in Rats"

_nutrients, 2021, doi:10.3390/nu13072128_

Round 1

Reviewer 1 Report

Martin and co-authors describe the effects of calorie restriction on different metabolic parameters. Although the concept was simple and straightforward, there are some major issues that needs to be addressed.

  1. Abstract: The conclusion needs to be revisited. Without proper comparison between N and HR, the conclusion is based on visual evaluation and not statistics. Although Npy and Pomc are indeed comparable between N and HR, the overall trend of the data does not reflect the well-reported effects of CR on these neuropeptides.
  1. Introduction: Clinically, there is no reason why a human who is trying to lose weight will still consume high fat diet instead of a low-calorie diet which can be coupled with calorie restriction. So, what is the premise of using HFD with calorie restriction in this experiment what is actually happening clinically?
  1. Materials and Methods. The goal of the experiment, as also reflected in the title, is to provide evidence that calorie restriction reverses the effect of adlib feeding. However, the major concern is that the rats on HR were never obese. Thus, I will contend on the use of the word reverse in the title. Statistics. Can the authors provide a direct comparison among 4 groups in all datasets? The current analysis provides insights on the effect of diet, restriction and their interaction. It would be interesting the directly compare N with HR as they the same BW but different diet and level of restriction.
  1. Results: Please provide the absolute food intake data both in grams and calorie, number of replicates should be provided either as scatter plots or in the legend.
  1. Discussion: Given the strong phenotype on both weight and food intake, it is surprising that the effects on Agrp and Npy are different. Similar observations can be found with Pomc and Cart. Can the authors provide an explanation why restriction and diet have a different effect in Npy and Agrp, same with Pomc and Cart, given that each pair have similar effects on feeding?

Reviewer 2 Report

The authors analyzed the effects of restriction of high-fat diet (HFD) on weight loss, circulating gut hormone levels, and expression of hypothalamic neuropeptides and found that such restriction normalizes the neuropeptide Y, the agouti-related peptide, and the anorexigenic factors proopiomelanocortin expression, as well as ghrelin and leptin levels.

Overall, a well-organized, well-written, and nice manuscript, addressing an interesting topic that still needs further investigation. I have no comments.

Author Response

We are very grateful for the comment of the Reviewer, which represents an extraordinary encouragement for our work.

Reviewer 3 Report

 Obesity is a world wild concern, which  increases the risk of other diseases and health problems.  There are many various dietary approaches to weight loss., however there is a need  to develop more effective strategies. 

The article entiltled :"Caloric restriction reverses the metabolic dysfunction and the 2 changes in hypothalamic neuropeptides associated with obesity 3 independently of the dietary fat content in rats" shows the data from in vivo study, which support the concept that restriction of a HFD may be used as an initial therapy in overweight and obese patients.  It seems to be very interesting aprouch.  I agree with the authors that the development of first step dietary interventions not involving a dramatic change in the habitual diet of overweight or obese people could represent an interesting alternative in the initial stages of obesity treatment. 

The research methodology is well written, the data are clear and well presented.

Author Response

(The authors gave the same response as above.)
